# The Role of MMPs in the Era of CFTR Modulators: An Additional Target for Cystic Fibrosis Patients?

**DOI:** 10.3390/biom13020350

**Published:** 2023-02-10

**Authors:** Renata Esposito, Davida Mirra, Giuseppe Spaziano, Francesca Panico, Luca Gallelli, Bruno D’Agostino

**Affiliations:** 1Department of Environmental Biological and Pharmaceutical Sciences and Technologies, University of Campania “Luigi Vanvitelli”, 81100 Caserta, Italy; 2Department of Health Sciences, University “Magna Graecia” of Catanzaro, 88100 Catanzaro, Italy

**Keywords:** cystic fibrosis, matrix metalloproteases, lung remodeling

## Abstract

Cystic fibrosis (CF) is a high-prevalence disease characterized by significant lung remodeling, responsible for high morbidity and mortality worldwide. The lung structural changes are partly due to proteolytic activity associated with inflammatory cells such as neutrophils and macrophages. Matrix metalloproteases (MMPs) are the major proteases involved in CF, and recent literature data focused on their potential role in the pathogenesis of the disease. In fact, an imbalance of proteases and antiproteases was observed in CF patients, resulting in dysfunction of protease activity and loss of lung homeostasis. Currently, many steps forward have been moved in the field of pharmacological treatment with the recent introduction of triple-combination therapy targeting the CFTR channel. Despite CFTR modulator therapy potentially being effective in up to 90% of patients with CF, there are still patients who are not eligible for the available therapies. Here, we introduce experimental drugs to provide updates on therapy evolution regarding a proportion of CF non-responder patients to current treatment, and we summarize the role of MMPs in pathogenesis and as future therapeutic targets of CF.

## 1. Introduction

Cystic fibrosis (CF) is an autosomal recessive disorder and the most common life-limiting fatal disease in the Caucasian population, affecting approximately 100,000 individuals around the world and over 42,000 in Europe [1]. According to the CF Foundation 2018 Registry Report, the rate of survival for CF patients is 47.4 years (people born in 2018 in the United States) [2]. The mutation characterizing the disease affects the CF gene encoding a protein called “the cystic fibrosis transmembrane conductance regulator” (CFTR) [3]. CFTR is a chloride channel included in the adenosine triphosphate (ATP)-binding cassette (ABC) transporter family, whose primary function is mediating the passage of chloride ions and other electrolytes from inside to outside the epithelial cells on which it is expressed [4]. Once the epithelial surface is reached, CFTR acts synergistically with other ion channels such as epithelial sodium channels (ENaCs). Notably, in the lungs, CFTR controls the chloride and bicarbonate ion flux across the epithelium into the airway surface liquid (ASL), playing a crucial role in regulating the ASL pH and protein composition [5]. Mutations in the CF gene result in CFTR variants affecting expression or activity. According to the last classification, CFTR mutations are clustered into seven classes, and the most common mutation F508del is the prototype of Class II which concerns approximately two-thirds of all CFTR alleles in CF patients. F508del can also be included in class III with a channel-gating defect, as well as in class VI, showing an accelerated turnover at the apical side of airway epithelial cells [6,7]. Regardless of mutations, CF patients experience a loss of bicarbonate secretion in airways and hyperabsorption of Na^+^ from the ASL, which induces acidification, dehydration, volume depletion, and increased viscosity of the mucus [8]. This microenvironment promotes airway inflammation with the activation of airway epithelial cells and fibroblasts, progressive thickening of the mucus layer, and impaired mucociliary clearance [9,10]. Moreover, major bacterial and viral colonization and infection happen, thus contributing to self-perpetuating and chronic inflammation and airway injury [11,12]. However, CF variants influence other organs with secretory functions such as the gastrointestinal system and pancreas; however, progressive destruction of the pulmonary tissue remains the major cause of morbidity and mortality in CF subjects [13]. Immune system imbalance was observed in CF pathogenesis, with the presence of enormous activated phagocytes without efficient antibacterial activity, as well as leukocyte and neutrophil infiltration into the lungs [14]. Thus, *Haemophilus influenzae*, *Pseudomonas aeruginosa*, and *Staphylococcus aureus* easily flourish in a dysfunctional airway microenvironment, leading to persistent infection, which results in toxic product releases such as proteases, reactive oxygen species, and cytokines [15]. In fact, abnormal levels of inflammatory and remodeling mediators have been found in CF patient airways [16]. Consequently, further immune cell recruitment promotes progressive lung tissue remodeling, a phenomenon in response to aberrant attempts to repair injured tissue, characterized by hyperplasia of goblet and basal cells, squamous metaplasia, and an increase in epithelial height [17]. Hence, the integrity of lung tissue is lost and replaced by damaged structural cells. These changes promote progressive impaired lung function, hypoxemia, and bronchiectasis associated with pulmonary symptom onset and ventilation/perfusion mismatch in the CF lungs. They result in so-called acute pulmonary exacerbations (APEs), which in turn contribute to CF morbidity and mortality [18]. Interestingly, lung remodeling is already observed in the early life of both small and large CF patient airways [19,20]. In fact, increased activity related to proteases, including matrix metalloproteinases (MMPs), is already seen during childhood [10]; hence, an early diagnosis of the disease is essential, as is an early start of treatment [21]. Like a systemic disease, the pharmacological treatment of CF is very complex, and several drugs have been approved, in addition to symptomatic therapy with inhaled antibiotics, airway clearance techniques, pancreatic enzyme replacement, and nutritional support [22,23]. Currently, four different groups of CFTR modulators are commercially available (potentiators, correctors, stabilizers, and amplifiers), transforming CF from a life-limiting to a lifelong chronic disease. Notably, ivafactor is the first licensed modulator by the regulatory authorities [24,25] based on real-world efficacy with a 50% reduction in pulmonary exacerbation in patients who did not show improvements in FEV1 at the beginning of therapy [26]. Furthermore, a combination of drugs with different actions was introduced to maximize efficacy. In fact, a triple combination of elexacaftor/tezacaftor/ivacaftor results in more efficacy for patients with the F508del mutation, maximizing the rescue of CFTR function [27,28,29,30]. Unfortunately, despite the steps forward in the treatment of CF, identifying new targets and drugs remains a challenge. Dysregulated MMP activity has been linked to the pathogenesis of numerous chronic lung diseases, including asthma, emphysema, and acute lung injury [31,32]. In fact, abnormal remodeling of tissue associated with the accumulation of extracellular matrix (ECM) is a hallmark of these diseases in which MMPs play a key role. However, more recently, MMPs have been shown to be linked to lung remodeling, a key driver of lung severity in CF patients, but there are not yet any specific antifibrotic treatments [33]. Taking these data into account and given the lack of a therapeutic approach useful for all CF patients, in this review, we discuss the link between MMPs and CF, as well as their potential role as future targets for CF individuals, Figure 1.

## 2. MMPs in CF Pathogenesis

One of the most important features of CF onset is progressive lung remodeling promoted by a significant increase in protease activity. In fact, CF patients have an imbalance between proteases and protease inhibitors, required physiologically for the equilibration of defense mechanisms and prevention of tissue damage [34]. The alteration of these dynamic network leads to proteolytic activity dysfunction in the lung as reported by many studies that brought to light changes in protease levels, notably in MMPs [35]. 

Overall, more than 20 MMPs have been described as key drivers of lung remodeling [36], and we report those with significant dysfunctions related to CF onset and progression, whose modulation could represent an important step forward in diagnosis, monitoring, and additional therapy for better management of the disease. 

MMPs are a superfamily of metallo-endopeptidases known as metzincins, synthesized as inactive proenzymes by several structural and immune cells such as macrophages, neutrophils, epithelial cells, endothelial cells, and fibroblasts [36]. While polymorphonucleates produce MMPs in a constitutive manner, many other cells release MMPs only after an inflammatory trigger and stimulation related to tissue remodeling and wound repair by transcription factor modulation [37]. Once synthesized, they became active by specific cleavage and, in turn, can cleave targeted substrates to perform biological functions. In fact, they contribute to the destruction of connective tissue and the alveolar epithelium, the release of cytokines and growth factors, and the regulation of cell mobility and migration by extracellular matrix (ECM) remodeling [38]. Moreover, they are involved in the wound repair process thanks to their capacity to catalyze the normal turnover of the ECM. In general, elevated MMP activity results in harm to lung homeostasis with low ECM turnover associated with an impaired repair phenomenon or excessive ECM accumulation associated with tissue fibrosis [39]. Moreover, MMPs can influence CFTR and ENAC channel structures via proteolytic breakdown, contributing to CF pathogenesis and progression [40]. Regardless of the molecular mechanism via which MMPs can promote CF onset, their role in the disease is supported by huge studies. For example, MMP-1, MMP-8, and MMP-9 levels were found higher in CF patients than in healthy controls [41]. Interestingly, a major increase in MMP-1 was observed during symptom exacerbations, indicating MMPs as potential additive biomarkers of disease severity. In contrast, other studies suggested a protective role of MMP-10 against bacterial infection because of its ability to modulate macrophage inflammation. However, the changes in MMP levels are often correlated to changes in MMP inhibitors such as α2-macroglobulin and the tissue inhibitors of metalloproteinases (TIMPs) [42]. A relationship between MMP and bacterial infection in CF subjects has also been found. Lastly, neutrophil elastase (NE) activity was found to be higher in CF childhood BAL, and its levels correlated with FEV1 [43]. Overall, MMPs can exert both beneficial and deleterious effects depending on the cellular source and the disease stage, Figure 2.

**MMP-9** represents the most abundant MMP in bronchopulmonary secretions derived from CF patients. MMP-9 is constitutively expressed by neutrophils, which are the main source of MMP-9 in neutrophilic inflammation characterizing CF disease. After inflammatory stimuli, other cellular types can produce MMP-9, such as macrophages and epithelial cells [44]. Once synthesized as a pro-enzyme, MMP-9 is further transformed into an active enzyme through the loss of its pro-domain. Many observational studies detected an association between BAL MMP-9 levels and CF progression, identifying elevated quantity and activity in the lower-airway secretions of CF patients [45,46,47]. Notably, BAL MMP-9 expression was related to CF severity in children with CF, pointing to its role in the earlier phase of the disease [48]. The role of NE as a key driver of MMP-9 activity is noteworthy, and researchers have referred to a causal link between the two proteases in CF as surrogated by a direct correlation between the increase in MMP-9 and NE activity [49]. Unlike NE, TIMP-1 is an endogenous inhibitor of MMP-9 activity, and a correlation between MMP-9 and TIMP-1 activity has been also detected [50]. In fact, an increase in the MMP-9/TIMP-1 ratio has been reported in the sputum and BAL of CF children and adults. The role of MMP-9 in the amplification of airway inflammation and associated lung tissue damage is supported by its ability to cleave and activate proinflammatory mediators such as IL-1-β and IL-8 release by inflammatory cells recruited into airways [51]. In turn, the proinflammatory transcription factor AP-1 is probably responsible for the increased expression of MMP-9 during inflammation. Moreover, MMP-9 levels were higher during CF progression, characterized by an acute pulmonary exacerbation associated with higher inflammation requiring systemic antibiotic treatment [52]. Overall, MMP-9 levels are positively correlated with the degradation of basement membrane collagen, decline in lung function, and onset of bronchiectasis in CF patients. Moreover, another substrate that can be activated by MMP-9 is serum surfactant (SP-D), an innate defense lectin secreted in the lungs. Previous studies identified serum SP-D as a potential biomarker of CF according to its adverse relationship with the FEV-1 parameter [53]. Therefore, the link between elevated levels of MMP-9 and SP-D is useful for supporting the role of MMP in CF pathogenesis. A recent study investigated both the levels and the activity of MMP-9, showing that blood MMP-9 activity is negatively associated with FEV-1, and that MMP-9 protein is positively associated with *Staphylococcus aureus* and *Pseudomonas aeruginosa* infections in CF. The significant correlation between FEV1 decline and MMP-9 increase could make MMP a useful additive biomarker of disease progression. However, the inhibition of MMP-9 alone did not affect goblet cell metaplasia, mucin secretion, and the emphysema onset in an animal study but reduced the bronchial obstruction by enhancing mucus clearance [54].

**MMP-12** is produced by macrophages following an inflammatory trigger in CF airways. MMP-12 is known as macrophage elastase for its elastolytic capacity, and its function can be related to the migration of macrophages toward airway tissues [55]. Recently, the expression of MMP-12 in CF individuals has been demonstrated, and the association between MMP-12 increase and lung destruction has been reported [49,56]. Interestingly, the rs2276109 polymorphism, located in the MMP-12 promoter, was identified, and the decrease in MMP-12 expression linked to polymorphism was positively associated with the FEV1 percentage predicted in patients with CF [57]. To better understand the function of MMP-12 in CF pathogenesis, a βENaC-Tg mouse was used as an experimental animal model mimicking the CF disease, and the inhibition of MMP-12 significantly reduced the emphysema-like features. Taken together, these studies give interesting evidence that the proteolytic activity of MMP-12 may contribute to the pathogenesis of structural lung damage and lung function decline in patients with CF [56,58]. 

**MMP-2** is expressed constitutively in lung cells and is involved in several biological functions such as inflammation and angiogenesis. Hence, MMP2 can boost airway injury by promoting abnormal remodeling and an impaired immune response to the infection in CF patients [59]. Interestingly, it seems able to influence CFTR activity, as reported by in vitro studies that describe a chloride flux increase after MMP-2 inhibition, supporting its involvement in ASL dehydration and increased bacterial colonization [60]. Otherwise, a recent paper found significantly low blood MMP-2 levels versus healthy subjects and even lower levels during acute CF exacerbation [61]. The authors speculated that the decreased expression of MMP-2 results from the binding site absence of proinflammatory transcription factors in the MMP-2 gene, which could hinder its activation by proinflammatory agents. Lastly, an upregulation of MMP-2 has been found during wound repair [62]. In fact, MMP-2 seems to be involved in epithelial–mesenchymal transition, in which airway epithelial cells start to produce collagen deposition [59]. However, further studies will be useful to elucidate the role of MMP-2 in CF onset and evolution.

**MMP-7** is produced in constitutive manner by airway epithelial cells, and its significant increase has been observed in CF individuals [40]. Notably, MMP-7 levels were increased in migrating airway epithelial cells during human wound repair. Therefore, like other MMPs, MMP-7 can also modulate airway re-epithelialization, inflammation, host defense, and cell growth, playing a critical role in the injury lung response in CF patients [63,64]. 

**MMP-8** is a neutrophil collagenase with several biological effects regarding the modulation of cytokines, immune activity, and repair process [65]. MMP-8 concentration has also been found augmented in the airway secretions and blood of patients with CF. Not strikingly, MMP-8 expression was significantly correlated with lung function decline (MMP-8 level vs. %FEV1, r = −0.468, *p* < 0.001) [61]. 

## 3. Experimental Drugs in CF Treatment

Research on new therapeutic approaches is ongoing [66]. For example, the safety and pharmacokinetics of three new compounds (nesolifactor, dirofactor, and posenofactor) were investigated in phase I–II clinical trials (NCT03500263/NCT03251092/NCT03591094) in individuals with CF, with an acceptable tolerability profile and effectiveness related to the early results. Promising results from these trials would increase the clinical efficacy of existing modulators and provide a good therapeutic option for patients whose genotype does not respond to current therapies [67,68]. In fact, there is a non-negligible proportion of CF patients (more than 10%) who are non-responders to CFTR-modulating drugs. Notably, for patients harboring mutations that result in no production of CFTR protein due to nonsense mutations, premature stop codons, and canonical splice or deletion mutations, gene therapy is gaining ground. Hence, ongoing efforts to develop readthrough agents for nonsense mutations are ongoing. For example, small-molecule inhibitors of the nonsense-mediated mRNA decay (NMD) pathway such as SMG1 inhibitor (SMGi) can improve CFTR transcript production and stability in cells harboring W1282X CFTR, providing potentially better clinical benefits [69,70]. Lastly, for faulty genes with mutations resulting in a premature termination codon in the CFTR mRNA, readthrough agents such as ELX-02 could represent gene-editing tools into personalized approaches for CF therapy. The rationale for using the aminoglycoside analog ELX-02 (eukaryotic ribosomal selective glycoside) is based on its ability to induce translational readthrough by binding to the decoding site in the small subunit of the ribosome. Therefore, the probability of the full-length protein being produced increases. ELX-02 also decreases NMD and increases the steady-state pool of transcripts available to produce more proteins [71,72]. A phase II clinical trial (NCT04135495) assessed the safety and efficacy of ELX-02 escalating doses with or without ivacaftor in 16 CF patients with at least one G542X allele. Positive results could provide the opportunity to introduce, in CF therapy, agents able to suppress the CFTR nonsense mutation for enhancing CFTR activity. Overall, gene-editing tools could represent a strategy to achieve permanent correction by replacing the CFTR faulty gene with a healthy gene. In this scenario, several other strategies are being developed including RNA-based strategies such as mRNA, t-RNA, and antisense oligonucleotides [73,74,75,76]; however, fewer clinical trials for CF RNA-based therapy exist. Nevertheless, the introduction of the correct version of CFTR mRNA by the delivery system ensures its use not only for nonsense mutations, but for all types of variants.

## 4. Therapeutic Agents Targeting MMPS

Despite many steps forward in the treatment of CF, resolutive therapy does not yet exist. Scientific research into new targets is an attractive challenge. In this context, modulating MMP activity is becoming increasingly interesting, as they are potential targets for the treatment of CF. In fact, the potential benefits of targeting impaired MMP activity in CF have led to several randomized clinical trials to investigate if MMP levels are modulated by CF conventional therapies. 

The development of anti-MMP therapeutics has not always led to satisfactory results due to poor safety or lack of efficacious drug delivery in the lungs. In fact, the current MMP inhibitors available are not specific, have a poor oral bioavailability, and induce several adverse reactions [77]. The inhibition of selective MMP could represent a better therapeutic strategy. Several compounds have been screened, and small, stable molecules have been selected with a high affinity and specificity. For example, nanobodies for MMP-8 inhibition were investigated in mouse inflammatory models with a micromolar IC_50_ value for inhibition activity (Table 1) [78]. Further, andecaliximab, a monoclonal antibody against MMP-9, was developed at Gilead, which selectively binds to the allosteric site of MMP-9, reducing fibrosis in patients with idiopathic pulmonary fibrosis, particularly in patients with elevated levels of IFN-gamma (Table 1) [79]. Interestingly, the common use of several classes of antibiotics [80,81,82] in CF patients prompted a recent phase II clinical trial (NCT01112059) to assess the capacity of antibiotic tetracyclines in reducing MMP levels in CF inpatients harboring the F508del mutation and receiving standard of care by analyzing the mechanism to restore the protease–antiprotease imbalance. The results showed that the use of doxycycline in addition to standard therapy reduced MMP-9 levels and activity, as well as inflammatory markers, in CF inpatients compared to those who did not receive doxycycline [83]. Moreover, in the same clinical trial, an increase in TIMP-1 levels was detected, showing the ability of doxycycline to restore MMP-9/TIMP-1 balance via MMP-9 neutralization and TIMP-1 enhancement (Table 1). Lastly, other papers described that doxycycline harbors anti-MMP activity, also affecting MMP-9 transcription or inhibiting its synthesis from human endothelial cells [84,85]. In addition to doxycycline, the antiprotease activity was associated with other antibiotic classes such as aminoglycoside, cephalosporin, carbapenem, and quinolones [86]. Moreover, azithromycin therapy has been associated with increased secretion of MMP-9 by alveolar M2 macrophages. In fact, CF patients receiving azithromycin have higher MMP-9 levels, as well as alveolar macrophage polarization toward the M2 subtype, thus promoting clinical benefits through a faster ECM turnover (Table 1) [87,88,89,90]. Another therapeutic agent able to reduce MMP levels is the recombinant human deoxyribonuclease (Dornase-Alpha), the only mucus-degrading agent able to reduce mucus viscosity through an inhaled once daily formulation [91]. Interestingly, MMP-8 and MMP-9 concentrations in CF BAL were reduced following its administration. Furthermore, Dornase-Alpha can modulate NE activity in CF sputum, confirming its ability to influence the protease–antiprotease balance (Table 1) [41,42,43,44,45,46,47,48,49,50,51,52,53,54,55,56,57,58,59,60,61,62,63,64,65,66,67,68,69,70,71,72,73,74,75,76,77,78,79,80,81,82,83,84,85,86,87,88,89,90,91,92]. Lastly, MMP activity is fine-tuned by several endogenous factors. Of recent interest is the role of an MMP-2 and MMP-9 inhibitor, called human epididymis protein 4 (HE4), expressed in the airway epithelium. The literature has reported a significant increase in HE4 in CF patients associated with lung function decline, airway fibrosis, and inflammation (Table 1) [93]. Moreover, high expression of HE4 promoted fibrosis by inhibiting the activities of MMPs in renal fibrosis [94]. Overall, abnormal HE4 expression is directly linked to CFTR dysfunction. Interestingly, a recent paper showed how HE4 levels and resulting inflammation could be reduced by lumacaftor/ivacaftor or tezafactor/ivafactor treatment through NF-kB inhibition. Hence, we speculate that MMP activity could be, in turn, influenced by this treatment [95], promoting a major ECM degradation able to counteract the fibrosis phenomenon. These data highlight the importance of MMP fine-tuning, being able to exert both beneficial and deleterious effects depending on the cellular source and the disease stage. 

## 5. Conclusions

CF presents a remarkable dysregulation of the protease/antiprotease balance, and MMPs have been implicated in the onset and severity of the disease. Their inhibition could be an interesting approach to counteract the downstream impact of protease dysfunction, slowing the progression of lung damage and bronchiectasis. Interestingly, the specific inhibition of MMPs represents a potentially more efficacious strategy, preserving the beneficial activities of MMPs. Therefore, this approach leads to new insight into the complex molecular basis of protease activity, and characterizing the protease activity on cells isolated from patients could help in diagnosis, prognosis, and pharmacological treatment. To date, treatment options for CF are mutation-dependent, and no viable options exist to address all CF patients. In fact, many CF patients without effective care still exist. In conclusion, the most important challenge will be to find effective therapies for all patients, changing CF into a disease that is no longer life-limiting by improving and delivering better healthcare. In this scenario, antiprotease strategies could be relevant to limit tissue damage in CF lung disease, in addition to CFTR modulators. In the future, it could be interesting to determine whether MMPs are able to contribute to CFTR regulation via proteolytic degradation, supporting the importance of MMP inhibitors for CF therapy.

## Figures and Tables

**Figure 1 biomolecules-13-00350-f001:**
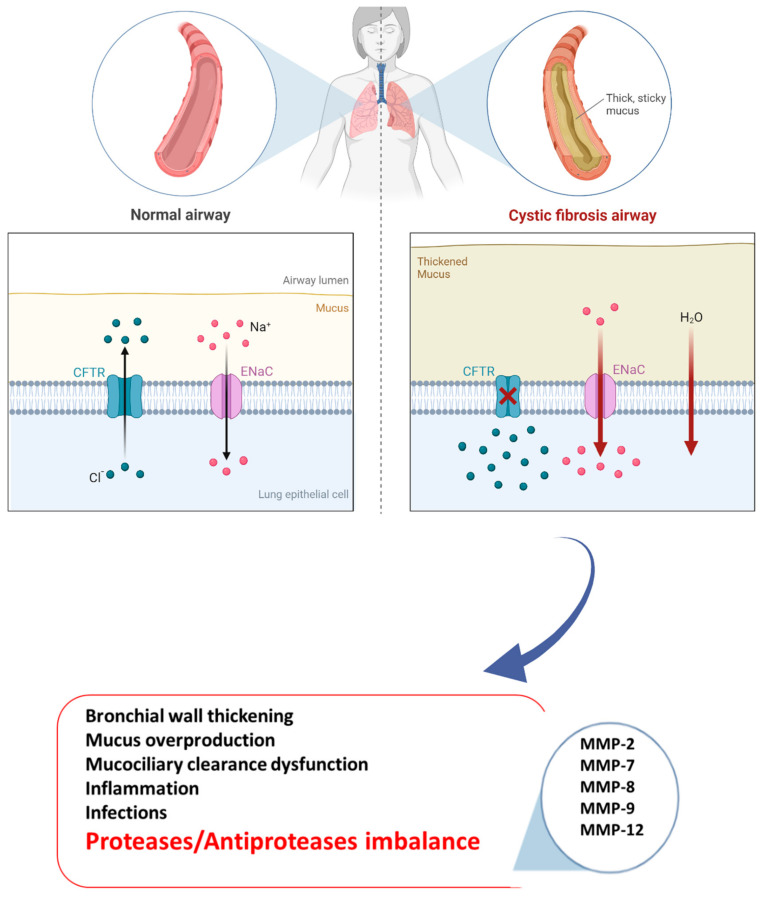
Pathogenesis of CF disease.

**Figure 2 biomolecules-13-00350-f002:**
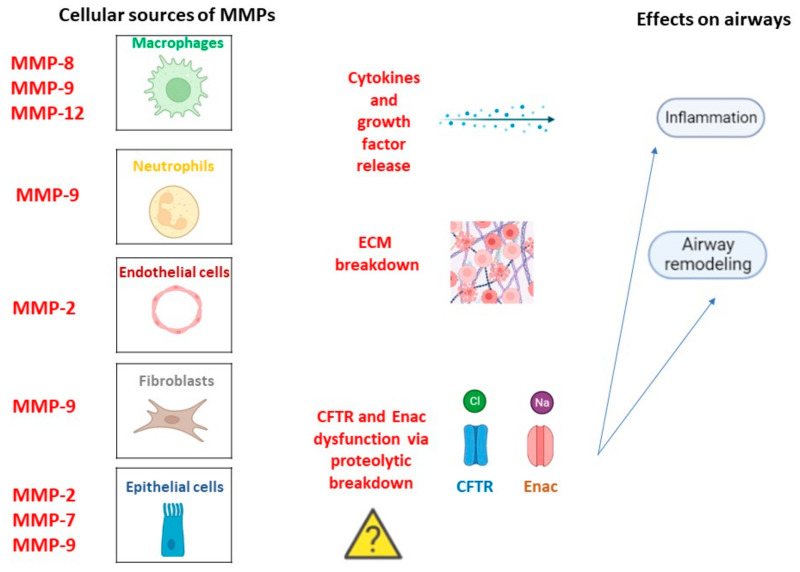
MMPs in CF pathogenesis. Cellular types involved in MMPs release in a constitutive manner or following inflammation trigger and related lung homeostasis dysfunctions.

**Table 1 biomolecules-13-00350-t001:** Therapeutic agents targeting MMPs. Red arrows indicate increase or decrease.

Drug	MMPs Targeting	Mechanism of Action	Effects	Experimental Phase	Ref.
**MMP-8 inhibitors (nanobodies)**	MMP-8	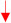 MMP-8	Major protection during systemic inflammation	Animal study	[78]
**Andecaliximab**	MMP-9	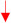 MMP-9	Reduced expression of ECM components through a decrease in TGF-β signaling	Animal study	[79]
**Doxycycline**	MMP-9	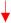 MMP-9 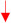 inflammatory markers 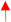 TIMP-1	FEV1% improvement andreduction of pulmonary exacerbation	Phase IINCT01112059	[83]
**Azithromycin**	MMP-9	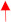 MMP-9 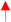 polarization of macrophages toward M2 subtype	Faster ECM turnoverReduction in neutrophil counts and serum inflammatory markers	Phase IVNCT00431964	[87,88]
**Dornase-Alpha**	MMP-8MMP-9	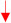 MMP-8 and MMP-9	Less mucus viscosityLung function improvement	Phase IVNCT00680316	[91,92]
**CFTR modulators**	MMP-2MMP-9	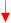 HE4	MMP-2 and MMP-9 increaseMajor ECM degradation	Observational study	[93,94,95]

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
