# Peer review of "The Role of MMPs in the Era of CFTR Modulators: An Additional Target for Cystic Fibrosis Patients?"

_biomolecules, 2023, doi:10.3390/biom13020350_

Round 1

Reviewer 1 Report

Esposito and colleagues have choosen to review the topic "The role of MMPs in the era of CFTR modulators: a new addi-2 tional target for cystic fibrosis patients?", which is a timely topic that could serve to summarise detailed findings on MMPs in one out of many more lung diseases for which protease/antiprotease imbalance plays a role.

The manuscript as such compiles an impressive number of 155 references, demonstrating  that the autors thoroughly researched the literature. But: the content provided does not match the title in about half of the manuscript. A lot of text is dedicated to CF therapeutics, thus, the information on MMP therapeutics is rather lost between the whealth of information on other therapuetic strategies in CF. Especially the chapter "MMPs in CF pathogenesis" is very useful to read for those who do not want to go through a set of publications accumulated for now more than a decade. 

The reviewer strongly suggests to shorten the manuscript considerably in order to provide a specialized review for the audience that is detailed on MMPs, but does not attempt to review e.g. the current status on CF modulator therapies (which will be a moving target anyway as more compounds are developed).

Specifically:

- Figure 1 is not informative on MMPs, 95% are unrelated to MMPs. On the other side, some importnat information on MMPs is not displayed, e.g. cell types involved in MMP production or consequences of MMP overexpression such as degradation of ECM

- line 78 to 81: the introduction of CFTR modulators in not state of the art. as the authours have compiled actual information in a later chapter, maybe they can write one or two phrases on CF therapy here in the introduction 

- line 84: rightfully, auttors state that MMPs are vital for pathogenesis in many other lung diseases. Thus, focus of the review to CF should be "MMP in CF" but not "CF pathogenesis" and "CF therapy" and "MMP-directed therapy" - an extension of the current content would be to describe whether or not MMPs have roles in other diseases distinct to their role in CF

-    the chapter PHARMACOLOGICAL TREATMENT FOR CF DISEASE belongs to a different manuscript and is not related to the topic choosen by the authors

- the chapter EXPERIMENTAL DRUGS IN CF TREATMENT belongs to a different manuscript and is not related to the topic choosen by the authors

- the chapter THERAPEUTIC AGENTS TARGETING MMPs is novel in this form and contributes to the filed 

- authours should check whether their references are correct - e.g. the sentence "However, CF variants influence other organs with secretory functions such as the gastrointestinal system and pancreas but progressive destruction of the pulmonary tissue remains the major cause of morbidity and mortality in CF subjects [17-18]." in line 55 to 57 links to references 17 and 18. While ref 17 appears to be correctly placed, reference 18 reports on  a different topic: Cappetta, D.; De Angelis, A.; Spaziano, G.; Tartaglione, G.; Piegari, E.; Esposito, G.; Ciuffreda, L.P.; Liparulo, A.; 558 Sgambato, M.; Russo, T.P.; Rossi, F.; Berrino, L.; Urbanek, K.; D'Agostino, B. Lung Mesenchymal Stem Cells 559 Ameliorate Elastase-Induced Damage in an Animal Model of Emphysema. Stem Cells Int 2018; 2018:9492038. doi: 560 10.1155/2018/9492038. PMID: 29731780; PMCID: PMC5872595.). Other titles that are rather unrelated to the topic of either MMPs or CF are: 11, 12, 15, 16, 18, ,22, 31, 32, 33, 34, 35, 36, 42, 43, 95, 122 (titles of these being:  Effects of simvastatin on cell viability and proinflammatory pathways in lung  adenocarcinoma cells exposed to hydrogen peroxide --- Leukotriene-mediated sex dimorphism in murine asthma-like features during allergen sensitization. --- Overview of Antiviral Drug Therapy for COVID-546 19: Where Do We Stand? --- Formulation of Solid Lipid Nanoparticles Loaded with Nociceptin/Orphanin FQ (N/OFQ) and Characterization in a Murine Model of Airway Hyperresponsiveness. --- Lung Mesenchymal Stem Cells Ameliorate Elastase-Induced Damage in an Animal Model of Emphysema. --- Circulating MicroRNAs Expression Profile in Lung Inflammation: A Preliminary Study. --- Activation of the nociceptin/orphanin FQ receptor reduces bronchoconstriction and microvascular leakage in a rabbit model of gastroesophageal reflux. --- Nociceptin inhibits airway microvascular leakage induced by HCl intra-oesophageal instillation. --- The involvement of sensory  neuropeptides in airway hyper-responsiveness in rabbits sensitized and challenged to Parietaria judaica. --- The 5-lipoxygenase inhibitor RF-22c potently suppresses leukotriene biosynthesis in cellulo and blocks bronchoconstriction and inflammation in vivo. --- Formulation and Characterization of Solid Lipid Nanoparticles Loading RF22-c, a Potent and Selective 5-LO Inhibitor, in a Monocrotaline-Induced Model of Pulmonary Hypertension. --- N/OFQ-NOP System and Airways --- Intratracheal Administration of Mesenchymal Stem Cells Modulates Tachykinin System, Suppresses Airway Remodeling and Reduces Airway Hyperresponsiveness in an Animal Model. --- Nociceptin/Orphanin Fq in inflammation and remodeling of the small airways in experimental model of airway hyperresponsiveness. --- Montelukast Improves Symptoms and Lung Function in Asthmatic Women Compared With Men. --- Disodium cromoglycate inhibits asthma-like features induced by sphingosine-1-phosphate. 

Author Response

RESPONSE TO REVIEWER 1 COMMENTS.

Esposito and colleagues have choosen to review the topic "The role of MMPs in the era of CFTR modulators: a new addi-2 tional target for cystic fibrosis patients?", which is a timely topic that could serve to summarise detailed findings on MMPs in one out of many more lung diseases for which protease/antiprotease imbalance plays a role.

The manuscript as such compiles an impressive number of 155 references, demonstrating  that the autors thoroughly researched the literature. But: the content provided does not match the title in about half of the manuscript. A lot of text is dedicated to CF therapeutics, thus, the information on MMP therapeutics is rather lost between the whealth of information on other therapuetic strategies in CF. Especially the chapter "MMPs in CF pathogenesis" is very useful to read for those who do not want to go through a set of publications accumulated for now more than a decade.

The reviewer strongly suggests to shorten the manuscript considerably in order to provide a specialized review for the audience that is detailed on MMPs, but does not attempt to review e.g. the current status on CF modulator therapies (which will be a moving target anyway as more compounds are developed).

Response: We thank the Reviewer for the sound suggestions and reorganized the manuscript according to the reviewers´ comments. We hope that our revised manuscript be better at showing the MMPs topic. In fact, we strongly reduced the manuscript body, focusing on MMPs function and their role as a possible drug target.

Specifically:

- Figure 1 is not informative on MMPs, 95% are unrelated to MMPs. On the other side, some important information on MMPs is not displayed, e.g. cell types involved in MMP production or consequences of MMP overexpression such as degradation of ECM

Response: We introduced a second more detailed figure (Figure 2) showing cellular types involved in MMPs production and the effects on the lung system related to their dysfunction.

- line 78 to 81: the introduction of CFTR modulators in not state of the art. as the authours have compiled actual information in a later chapter, maybe they can write one or two phrases on CF therapy

Response: Following the right observation of the Reviewer, we introduced a short description of CFTR modulators in the introduction.

- line 84: rightfully, authors state that MMPs are vital for pathogenesis in many other lung diseases. Thus, the focus of the review of CF should be "MMP in CF" but not "CF pathogenesis" and "CF therapy" and "MMP-directed therapy" - an extension of the current content would be to describe whether or not MMPs have roles in other diseases distinct to their role in CF.

Response: We introduced a brief sentence on the role of MMPs in other pulmonary diseases. 

-    the chapter PHARMACOLOGICAL TREATMENT FOR CF DISEASE belongs to a different manuscript and is not related to the topic chosen by the authors

Response: We removed the chapter entitled “PHARMACOLOGICAL TREATMENT FOR CF DISEASE” for focusing better on the real topic of the manuscript.

- the chapter EXPERIMENTAL DRUGS IN CF TREATMENT belongs to a different manuscript and is not related to the topic chosen by the authors

I removed the table since the topic is not the focus of manuscript.

Response: We significantly reduced the chapter “EXPERIMENTAL DRUGS IN CF TREATMENT”. We also removed Table 1 because it provided data not related to the main topic of the manuscript.

- the chapter THERAPEUTIC AGENTS TARGETING MMPs is novel in this form and contributes to the filed 

Response: We thank the Reviewer for the sound observation and we focused on this topic.

- authors should check whether their references are correct

Response: Based on the Reviewer’s suggestion, we checked all references by removing those that were not correctly placed. Finally, we reorganized the entire bibliography.

Reviewer 2 Report

Overall, this is an informative and useful review.

1. Some typos need to be corrected (e.g., Ivacaftor, lumacaftor, etc.).

2. The clinical trial of some investigated drugs could be discontinued (e.g., cavosonstat, Ataluren, etc). The authors need to show the current status.

3. Table 2 is too simple. The authors should summarize the specific effects of the MMP-related drugs in the table rather than just citing literature.

Author Response

RESPONSE TO REVIEWER 2 COMMENTS.

  1. Some typos need to be corrected (e.g., Ivacaftor, lumacaftor, etc.).

Response: We thank the Reviewer for the suggestion and we corrected typos in the manuscript.

  1. The clinical trial of some investigated drugs could be discontinued (e.g., cavosonstat, Ataluren, etc). The authors need to show the current status.

Response: We thank the Reviewer for the right observation. We checked the current status of the drugs mentioned in the manuscript. We reduced the paragraph, removing the description of the drugs whose results are not published yet. We also deleted Ataluren because its clinical development in CF was discontinued.

  1. Table 2 is too simple. The authors should summarize the specific effects of the MMP-related drugs in the table rather than just citing literature.

Response: We introduced a more detailed table.

Reviewer 3 Report

The authors provide and interesting overview of MMPs in the pathogenesis of CF airway disease.  It is a timely review in that symptomatic therapies are still needed to address inflammation in CF airways, particularly in relation to recurrent exacerbations.  Though studied in CF since the 1990s, interventions within the MMP pathway have not been a priority and interventions within this pathway could serve a valuable clinical purpose. 

Strengths

-       The review does an excellent job with background information and puts the objectives of the review in a clear and understandable context.

-       Defining the regulation and roles of specific MMPs is helpful to the reader and provides understanding to the complexity of the system in the airways.

Weaknesses

-        The discussion of the other therapies is not well incorporated into the theme of the review.  Review of modulator therapies is understandable and it is stated that not much is known how these therapies are impacting MMP expression and regulation.  However, the other secondary therapies such as mucolytics and bronchodilators are never discussed in context of MMPs and seem tangential.  If there is information as to how these other ancillary therapies are impacting MMPs, then it should be included.  Otherwise, these sections should be removed as they do not add to the understanding of MMPs in CF.

-       The role of section describing MMP-2 is certainly appropriate, but is not very clear and needs some editing.

-       The role of HE4 in CF is somewhat confusing.  HE4 expression is positively correlated with CF airway inflammation despite it being a MMP inhibitor.  At first glance, this relationship seems counterintuitive to the premise of the review, though certainly more complicated.  More clarification of this relationship would be interesting.

-       Some minor editing is needed

Overall this paper is an interesting and timely review of an important subject.  Some changes to increase focus and clarity would further improve the impact of the review. 

Author Response

RESPONSE TO REVIEWER 3 COMMENTS.

 Weaknesses

-        The discussion of the other therapies is not well incorporated into the theme of the review.  Review of modulator therapies is understandable and it is stated that not much is known how these therapies are impacting MMP expression and regulation.  However, the other secondary therapies such as mucolytics and bronchodilators are never discussed in context of MMPs and seem tangential.  If there is information as to how these other ancillary therapies are impacting MMPs, then it should be included.  Otherwise, these sections should be removed as they do not add to the understanding of MMPs in CF.

Response: We thank the Reviewer for the right observation. We checked in the current literature if symptomatic drugs such as mucolytics and bronchodilators could influence MMPs function in CF. There are not studies that describe a relationship between MMPs and the abovementioned drugs in the context of CF. Hence, we decided to remove their description from the manuscript.

-       The role of section describing MMP-2 is certainly appropriate, but is not very clear and needs some editing.

Response: Following the observation of the Reviewer, we explained clearer the role of MMP-2 in CF based on little information reported in the literature compared to other MMPs related to CF disease.

-       The role of HE4 in CF is somewhat confusing.  HE4 expression is positively correlated with CF airway inflammation despite it being a MMP inhibitor.  At first glance, this relationship seems counterintuitive to the premise of the review, though certainly more complicated.  More clarification of this relationship would be interesting.

Response: We extended the description of HE4 in a clearer form.  

-       Some minor editing is needed

Overall, this paper is an interesting and timely review of an important subject. Some changes to increase focus and clarity would further improve the impact of the review.

Response: We revised the entire manuscript to reorganize the manuscript for a better understanding of the topic.  

Round 2

Reviewer 1 Report

The reviewer thanks the authors for modifying the manuscript extensively - it is now focussed on MMP in CF and provides a very timely and useful compilation of the existing literature on that subject. As degradation processes of  lung tissue is a key process for many lung diseases, it willhopefully receive attentiin beyond the cystic fibrosis community.